# A Study of the Chemical Composition and Biological Activity of *Michelia macclurei* Dandy Heartwood: New Sources of Natural Antioxidants, Enzyme Inhibitors and Bacterial Inhibitors

**DOI:** 10.3390/ijms24097972

**Published:** 2023-04-28

**Authors:** Shixiang Chen, Bochen Wei, Yunlin Fu

**Affiliations:** 1College of Forestry, Guangxi University, Nanning 530004, China; shixiang_chen2022@163.com (S.C.); bochen0411@163.com (B.W.); 2Key Laboratory of National Forestry and Grassland Administration on Cultivation of Fast-Growing Timber in Central South China, College of Forestry, Guangxi University, Nanning 530004, China

**Keywords:** *Michelia macclurei* Dandy, heartwood, chemical composition, biological activity

## Abstract

The wood of *Michelia macclurei* Dandy (MD) is an excellent material that is widely used in the furniture, handicraft, and construction industries. However, less research has been conducted on the chemical composition and biological activity of heartwood, which is the main valuable part of the wood. This study aimed to investigate the chemical composition and biological activities of the heartwood of *Michelia macclurei* Dandy (MDHW) and to confirm the active ingredients. Triple quadrupole gas chromatography–mass spectrometry (GC-MS) was used to characterize the volatile components of MDHW, while ultra-performance liquid chromatography–mass spectrometry was used to analyze the non-volatile components (UPLC-MS). The total reducing power, 2,2-diphenyl-1-picrylhydrazyl (DPPH) radical, and 2,2′-azino-bis(3-ethylbenzothiazoline-6-sulphonic acid) (ABTS) radical scavenging assays, acetylcholinesterase and α-glucosidase inhibition assays, and an antimicrobial test of 4 gram bacteria were used to describe the in vitro bioactivities. The GC-MS analysis showed that the volatile components of MDHW were mainly fatty compounds and terpenoids, with sesquiterpenes and their derivatives dominating the terpene composition. β-elemene was the main terpene component in the steam distillation (11.88%) and ultrasonic extraction (8.2%) methods. A total of 67 compounds, comprising 45 alkaloids, 9 flavonoids, 6 lignans, and others, were found by UPLC-MS analysis. The primary structural kinds of the non-volatile components were 35 isoquinoline alkaloids. Alkaloids were the predominant active constituent in all MDHW extracts, including crude extracts, alkaloid fractions, and non-alkaloid fractions. These extracts all demonstrate some biological effects in terms of antioxidant, enzyme inhibition, and bacterial inhibition. The findings of this study show that MDHW is abundant in chemical structure types, has great bioactivity assessment, and has the potential to be used to create natural antioxidants, products that postpone Alzheimer’s disease and lower blood sugar levels and antibacterial agents.

## 1. Introduction

Natural products are organic molecules with a variety of structural properties and biological activities that have evolved via long-term natural selection and evolution. They are safer and more affordable than synthetic products [1]. Both the medications used in conventional medicine and the chemical pharmaceuticals that are frequently prescribed in modern medicine are either analogs or derivatives of natural chemicals. The biological and chemical makeup of many of the plants used in traditional medicine and folk cures is still insufficiently understood.

Free radicals have been linked to several illnesses, including atherosclerosis, cancer, diabetes, liver disease, and organismal aging, according to a recent pathological study [2]. Antioxidants not only scavenge free radicals but also inhibit their production. Alzheimer’s disease (AD) and diabetes mellitus (DM) are considered to be increasingly prevalent global public health problems in the 21st century [3]. As a result, medications for AD and DM have drawn a lot of interest. However, many synthetic antioxidants and medications that block AD and DM are inefficient and come with adverse side effects including nausea and diarrhea [4,5,6]. There is an increasing search for non-toxic and effective natural products to treat these diseases. Additionally, drug-resistant microorganisms are emerging and persisting at an increasing rate, indicating that the available conventional antibiotics are becoming less effective. As a result, it is crucial to discover new, potent antibacterial medications [7].

*Magnoliaceae* is the most primitive group of angiosperms and is widespread throughout the world. It is a traditional Chinese medicine used in Asia and North America. For example, the Chinese Pharmacopoeia has officially listed the *Magnolia officinalis cortex* and *Magnolia biondii* Pamp., which are rich in magnocurarine and volatile oil [8]. Alkaloids, lignans, minor levels of flavonoids, and steroids are all present in the non-volatile composition of *Magnoliaceae*, which is also known for being rich in aromatic compounds [9]. MD is a *Magnoliaceae* (*Michelia Linn*.) evergreen arbor plant, which is widely used in the furniture, handicrafts, musical instrument manufacturing, plywood, and construction industries, is an important native precious timber and multifunctional high-efficiency species in southern China [10]. Regrettably, investigations into MD’s medicinal value have rarely been reported. Instead, they have mostly concentrated on biomass, breeding, and the development and utilization of essential oils [11,12]. The MD chemical composition research is reflected in the analysis of the volatile oil components of flowers, leaves, and fruits, and it has been confirmed that these volatile oils have a variety of biological activities, which are better medicinal materials [13]. In Chinese folklore, the roots and bark have traditionally been used for the treatment of inflammation and heat-clearing [14], whereas the heartwood receives less attention than the other parts. A significant amount of secondary metabolites are accumulated in the heartwood, which is a part of the xylem. The heartwood’s characteristic odor, color, and natural durability, as well as its high economic worth, are caused by a significant buildup of secondary metabolites. For example, the heartwood of *Dalbergia odorifera* and *Santalum album* is ideal for crafting artistic crafts and sculptures as well as traditional Chinese medicine and aromatherapy [15,16]. The volatile components of MDHW are reported to be rich in sesquiterpenes such as β-elemene and β-bisabolene [17]. No studies have yet reported on the non-volatile components of MD.

Therefore, the purpose of this study was to carry out a comprehensive analysis of the chemical components of MDHW in terms of both volatile and non-volatile components using GC-MS and UPLC-MS to explore the biological potential of MDHW extracts, and to identify the main effective bioactive components. To determine the MDHW’s primary bioactive components, the antioxidant, enzyme inhibition, and antibacterial properties of the crude extract, alkaloid, and non-alkaloid fractions were assessed. This research aimed to provide guidelines for assessing the economic value and sensible application of MD. The study’s findings will provide new ideas for the development and utilization of MDHW in pharmaceutical, cosmetic, and nutraceutical products.

## 2. Results and Discussion

### 2.1. Volatile Components of MDHW Determined by GC-MS

In this study, two different extraction methods were used to analyze the volatile oil of MDHW using GC-MS. Table 1 displays the results of the top ten and the same essential oil components produced by steam distillation (SD) and ultrasonic extraction (UE), and Appendix A displays all essential oil components. Respectively, 81 and 115 peaks were detected by SD and UE, among which 59 and 38 compounds were identified, and 97.54% and 76.15% accounted for the peak area. According to the species, oxygenated sesquiterpenes (57.66%), sesquiterpene hydrocarbons (38.74%), monoterpene hydrocarbons (0.78%), and aromatic compounds (0.37%) made up the majority of the essential oil composition discovered by SD. Fatty compounds (59.93%) and sesquiterpene hydrocarbons (13.23%), as well as tiny quantities of aromatic compounds (2.38%), oxygenated sesquiterpenes (0.54%), and monoterpene hydrocarbons (0.07%) constituted the majority of the essential oils extracted from UE. These data indicate that the composition and content of the essential oils produced by these two extraction methods are significant differences.

UE extracts a wider variety of essential oils, but the terpenoid level was not very high. The most common procedure for extracting essential oils is SD, and the compounds detected are mainly terpenoids and their derivatives with high purity [18], as also confirmed by our results. For a preliminary investigation into plant essential oil constituents, UE is preferable since it is quick, simple to use, and can detect a wider variety of chemicals. SD is more suited for realistic production applications due to its high terpene extraction rate, low cost, and ease of use.

Twelve of the same components were extracted by both extraction methods (Figure 1), namely δ-3-carene, copaene, β-longipinene, β-elemene, sesquithujene, α-bergamotene, 1,9-aristoladiene, alloaromadendrene, α-bisabolene, cadina-1(10),4-diene, elemol, and guaiol. These compounds are structurally diverse and have been reported several times in the *Michelia Linn.* [9,19,20,21]. These results show that sesquiterpenes and their derivatives dominate the terpene composition of MDHW essential oil, while diterpenes and higher terpenes are rarely observed. This may be because *Magnoliaceae* is the most primitive angiosperm and therefore lacks or has lower levels of higher terpenes.

β-elemene was the highest terpenoid component in SD and UE (11.88% and 8.2%, respectively), which is consistent with that reported in the literature (the content of β-elemene is as high as 14.67%) [17], indicating that this compound could be used as a potential characteristic component of MDHW essential oil. In China, β-elemene is used as a class II non-cytotoxic anti-tumor medicine since it is primarily utilized in the treatment of many different types of malignant tumors without damaging liver or kidney function [22]. Guaiol (10.67%) was effective in inhibiting brown and white rot fungi [21], whilst β-eudesmol (9.68%) is one of the most studied and major biologically active sesquiterpenes with strong anti-tumor and anti-angiogenic activities [23].

According to the above analysis, MDHW has multiple types of essential oil constituents. Furthermore, major terpene constituents such as β-elemene and β-eudesmol have been used as clinical compounds, indicating that MDHW essential oil may be a new source for pharmaceuticals.

### 2.2. Non-Volatile Components of MDHW with UPLC-MS/MS

The non-volatile components of the ethanolic extract of MDHW were analyzed in this study using UPLC-MS, and the details of all the identified compounds are summarized in Table 2. A total of 67 compounds were identified, including 45 alkaloids, 6 lignans, 9 flavonoids, 3 terpenoids, 2 coumarins, and 2 lipids. To our knowledge, these non-volatile components have never been reported in previous MD research. Our results revealed that the major constituents of MDHW extracts were alkaloids. According to phytochemical results, MDHW extracts were abundant in different types of alkaloids, flavonoids, lignans, and coumarins. These compounds have been shown in numerous experiments to have antioxidant, antibacterial, anti-tumor, hypotensive, antiarrhythmic, and antidepressant functions [1], indicating that MDHW extracts may have a variety of bioactive properties.

Thirty-five isoquinoline alkaloids were discovered to be the primary types of MDHW alkaloid components in the current investigation. Therefore, based on the secondary fragments, the mass spectrometric cleavage pattern of isoquinoline alkaloids and relevant literature were referred to, the compound structures of MDHW isoquinoline alkaloids were identified and the proposed fragmentation pathways of the compounds were analyzed. Based on their basic heterocyclic nuclei, these isoquinoline alkaloids can be classified into six groups, viz., benzylisoquinoline alkaloids (compounds **4**, **12**, **14**, **15**, **18**, **32**, **37**, **39**, and **43**), aporphine alkaloids (compounds **3**, **6**, **7**, **8**, **10**, **11**, **13**, **16**, **17**, **20**, **24**, **27**, **31**, **33**, and **41**), protoberberine alkaloids (compounds **21** and **25**), tetrahydroprotoberberines alkaloids (compounds **2**, **19**, **34,** and **42**), protopine alkaloids (compounds **9** and **38**) and benzophenanthridine alkaloids (compounds **22**, **23,** and **26**)—the basic structures of which are shown in Figure 2. The MS/MS spectrum and proposed fragmentation pathways of 35 isoquinoline alkaloids are shown in Appendix A.

Most isoquinoline alkaloids are synthesized from benzyltetrahydroisoquinoline alkaloids, which can then be converted into tetrahydroproberberberine alkaloids and aporphine alkaloids, proberberberine is derived from tetrahydroproberberine alkaloids, protopine alkaloids is generated from tetrahydroproberberine alkaloids and is continuously metabolized to dihydrobenzophenanthridine alkaloids and benzophenanthridine alkaloids [24]. The MDHW alkaloids are dominated by simply structured aporphine alkaloids, which may be because MD belongs to the *Magnoliaceae*, which are the most primitive angiosperms.

#### 2.2.1. Benzylisoquinoline Alkaloids

Benzylisoquinoline alkaloids would lose their nitrogenous side chains and form fragmented peaks such as m/z 192, m/z 175, m/z 143, and m/z 137 [25]. The excimer ion peak in the positive ion mode of compound **4** was m/z 330 [M+H]^+^, and molecular formula was probably C_19_H_23_NO_4_. The fragment ions with higher abundance in the mass spectra all appeared below m/z 230, and the fragments with higher abundance were m/z 192 and m/z 137. These fragmentation behaviors are consistent with the mass spectral cleavage pattern of benzylisoquinoline alkaloids. m/z 299 was formed by the loss of the nitrogen-containing side chain [M+H-NH_2_CH_3_]^+^ from the parent structure. m/z 299 underwent β-cleavage to form m/z 175, and m/z 175 underwent further side chain breakage and loss of CH_3_OH fragments to form m/z 143. These fragmentation behaviors were consistent with those reported in the literature [26], thus, compound **4** was identified as (S)-reticuline. Interestingly, (S)-reticuline is an important intermediate compound in the metabolic pathway of many isoquinoline alkaloids, from which most benzylisoquinolines alkaloids, aporphine alkaloids, proberberberine alkaloids, and benzophenanthridine alkaloids are converted [27].

The excimer ion peak of compound **37** was m/z 314 [M+H]^+^, the possible molecular formula was C_19_H_23_NO_3_, and the presence of the fragments m/z 175, m/z 143, and m/z 137 in the mass spectra indicated that it was most likely a benzylisoquinoline alkaloid, and based on the existing reports of the chemical composition of *Magnocurarinaceae* [28], compounds **37** was identified as magnocurarine. The excimer ion peak of compound **32** was m/z 314 [M+H]^+^, which has a similar MS/MS spectrum to compound **37** (Appendix A), but both were isomers, so we presumed compound **32** to be armepavine based on the available literature [26]. The nitrogen atom on the parent structure of magnocurarine is attached to two methyls, while armepavine is attached to only one methyl. The two can be distinguished based on the pattern that benzylisoquinoline alkaloids readily lost NH_3_, CH_3_NH_2_, or (CH_3_)_2_NH. Magnocurarine lost (CH_3_)_2_NH to form the fragment m/z 269, and armepavine lost CH_3_NH_2_ to form the fragment m/z 283, and the breakage of the side chain would not form m/z 269. The chemical structure and proposed fragmentation pathways of compounds **4** and **37** are shown in Figure 3.

According to the similarity pattern, in combination with secondary fragments the and related literature [29], five benzylisoquinoline alkaloids were also observed in this study. Compounds **12**, **14**, **15**, **18**, **39**, and **43** were identified as norarmepavine, isococlaurine, (S)-coclaurine, O-methylarmepavine, laudanosine, and protosinomenine, respectively. Their MS/MS spectrum and possible fragmentation pathways are shown in Appendix A.

#### 2.2.2. Aporphine Alkaloids

The excimer ion peak of compound **3** was m/z 342 [M+H]^+^, and the possible molecular formula was C_20_H_24_NO_4_. The fragment ions with a higher abundance in the mass spectra all appeared above m/z 210, indicating that no cleavage of the parent structure occurred. Based on the cleavage pattern of isoquinoline alkaloids [25], compound **3** can be presumed to be benzophenanthridine or aporphine alkaloids. The fragment m/z 297 was formed when the parent ion loses nitrogen methyl [(CH_3_)_2_NH]^+^, after which m/z 265 was obtained by the continued loss of the side chain CH_3_OH to form a stable ternary oxygen ring, and then a molecule of CO was lost to form a five-membered ring of the parent structure to obtain m/z 237, and the side chain CH_3_CH of m/z 237 breaks to form m/z 209. Another cleavage pathway for m/z 297 was the loss of a molecule of methyl to form a carbonyl group to obtain m/z 282, continued with the loss of two hydroxyls to form a parent structurally stable five-membered ring to form m/z 265, followed by the loss of one molecule of CO to form a five-membered ring to form m/z 237, and finally the loss of the side chain CH_3_CH to form m/z 209. In summary, compound **3** was identified as magnoflorine in comparison with the literature [30]. 

The excimer Ion peak of compound **6** was m/z 342 [M+H]^+^, and the major fragment peaks and ion abundances in its mass spectra were similar to compound **3** (Appendix A), so they were isomers. Combined with the literature [29], compound **6** was identified as laurifoline. Due to the different positions of the hydroxyl on the D-ring, they can be distinguished by the presence of an ionic abundance of magnoflorine at m/z 219, while laurifoline was not. The chemical structure and proposed fragmentation pathways of compounds **3** and **6** are shown in Figure 4.

Based on the similarity pattern and combined with the accurate parent ion molecular weights, secondary mass spectrometry information, and relevant literature [29,31,32], compounds **7**, **8**, **10**, **11**, **13**, **16**, **17**, **20**, **24**, **27**, **31**, **33,** and **41** were identified as corytuberine, N, N-dimethylglaucine, bulbocapnine, apomorphine, N-methylasimilobine, asimilobine, ushinsunine, menisperine, anolobine, roemerine, xanthoplanine, michelalbine, and anonaine, and their chemical structures and proposed fragmentation pathways are shown in Appendix A. Among them, compounds **3**, **11**, **16**, **17**, **24**, **27**, **33**, and **41** have been isolated from *Magnoliaceae* [31,33,34]. 

#### 2.2.3. Protoberberine Alkaloids

The excimer ion peak of compound **21** was m/z 352 [M+H]^+^, and the possible molecular formula was C_21_H_21_NO_4_. After the successive losses of one molecule of methyl and one H, the parent structure formed a carbonyl and obtained the fragment m/z 336, immediately followed by the continued loss of one molecule of CO to form m/z 308. Another cleavage pathway was the formation of m/z 322 and m/z 294 after the successive loss of molecules of CH_3_ and CO. Finally, due to the presence of carbon–carbon single bonds at the C-5,6 positions, the parent structure lost 2 molecules of H to form a stable conjugate system and obtained m/z 292 with higher abundance. These followed the cleavage pattern of protoberberine alkaloids [25], and combined with the literature [33], compound **21** was identified as palmatine. 

The excimer ion peak of compound **25** was m/z 326 [M+H]^+^. m/z 321, m/z 307, and m/z 308 were formed by parent ions through the loss of one molecule of OH and H_2_O, respectively. m/z 309 formed a ternary oxygen ring after the loss of a methylenedioxy and successively lost a molecule of CO to form fragments m/z 279 and m/z 251. In addition, the parent ion can lose two hydrogens to form m/z 324. The parent ion can also lose both a molecule of CH_3_ and a molecule of OH to form fragmentation m/z 294, immediately followed by the continued loss of a molecule of CH_2_O to form the more stable fragmentation ion peak m/z 264. Combined with the cleavage pattern of protoberberine alkaloids [25], compound **25** was identified as cheilanthifoline. The chemical structure and proposed fragmentation pathways of compounds **21** and **25** are shown in Figure 5.

#### 2.2.4. Tetrahydroprotoberberine Alkaloids

Tetrahydroproberberine alkaloids frequently undergo a retro Diels–Alder (RDA) reaction at the C-ring, with fragments m/z 206, m/z 192, and m/z 165. These common fragments are used as diagnostic ions for the structural characterization of tetrahydroprotoberberine alkaloids [25]. In addition, if the substituent at the 13th position is methyl, the RDA reaction will most likely also occur in the B-ring.

The MS/MS spectra of compounds **2**, **19**, **34**, and **42** (Appendix A) showed similar RDA fragmentation behaviors. The main fragments of compound **2** were m/z 178 and m/z 151. m/z 178 and m/z 151 can be formed not only by RDA reaction through the B-ring of the parent structure but also by direct RDA reaction through the C-ring. Another characteristic ion, m/z 133, was formed by the loss of one molecule of water from m/z 151. Compound **2** was identified as (S)-scoulerine in light of the literature [26].

Similarly, m/z 192 and m/z 165 were the dominant ions in compound **42**, which can originate from both RDA cleavage occurring in the C-ring of the parent structure and from B-ring cleavage formation. m/z 192 further lost a molecule of methyl to form fragments m/z 177. To form a stable conjugated system, the parent structure compound **42** first lost two hydrogens and subsequently lost a molecule of methyl to form m/z 338. Compound 42 was identified as (S)-tetrahydropalmatine [33]. The chemical structure and proposed fragmentation pathways of compounds **2** and **42** are shown in Figure 6.

The C-ring on the parent structure of compound **19** underwent RDA reactions to form m/z 192 and m/z 179, the B-ring underwent RDA to form m/z 205 and m/z 165, and compound **19** was identified as corydaline [34]. Similarly, compound **34** was identified as tetrahydrocolumbamine [35]. The chemical structures and proposed fragmentation pathways of compounds **19** and **34** are shown in Appendix A.

#### 2.2.5. Protopine Alkaloids

The excimer ion peak of compound **9** was m/z 354 [M+H]^+^, and the possible molecular formula was C_20_H_19_NO_5_. The fragment ion with the higher abundance in the mass spectra was m/z 206, indicating that the parent structure underwent cleavage. Fragment m/z 206 was usually formed by the RDA cleavage of the alkaloid parent structure, and m/z 206 further lost a molecule of water to form fragment m/z 188. Another possible cleavage pathway for compound **9** was the loss of methyl from the parent structure to form m/z 339, followed by the removal of a molecule of H_2_O to form fragment m/z 336, removal of methylenedioxy to form a ternary ring, and finally the removal of a methyl group to form m/z 324. These above processes were consistent with the cleavage pattern of protopine alkaloids [25], and then by comparing with the literature [26], compound **9** was identified as protopine.

The excimer ion peak in the positive ion mode of compound **38** was m/z 370 [M+H]^+^, and the molecular formula was probably C_21_H_23_NO_5_. The parent structure lost one CH_3_ to form fragment m/z 355, followed by α-cleavage to obtain fragment m/z 190. The parent structure underwent RDA cleavage to form the fragments m/z 206 and m/z 165, and m/z 206 continued to lose a hydroxyl to form m/z 189. Compound **38** was identified as allocrytopine in combination with the literature [26]. The chemical structure and proposed fragmentation pathways of compounds **9** and **38** are shown in Figure 7.

#### 2.2.6. Benzophenanthridine Alkaloids

The fragments with higher abundance in compounds **22**, **23**, and **26** were all greater than m/z 200, indicating that the parent structure was not cleaved and therefore presumed to be benzophenanthridine alkaloids [25]. 

The excimer ion peak of compound **22** was m/z 354 [M+H]^+^, and the possible molecular formula was C_20_H_19_NO_5_. Based on the presence of m/z 337 and m/z 338, it can be presumed that the side chain of compound **22** contains hydroxyl, and these two fragments originated from the loss of hydroxyl or one molecule of water from a parent structure to form m/z 337, which continued with the loss of one molecule of methyl to form m/z 322. With the successive loss of CH_3_, H, and CH_2_O molecules from the parent structure, fragments m/z 338 and m/z 308 were formed, and compound **22** was identified as chelidonine [24,36]. 

The excimer ion peak of compound **23** was m/z 364 [M+H]^+^, and the possible molecular formula was C_22_H_21_NO_4_. The presence of methoxy in the parent ion was inferred from the fragment m/z 333 [M+H-CH_3_OH]^+^. m/z 349 was formed by the loss of one molecule of methyl from the parent structure, and then, it was presumed that the parent structure contained methylenedioxy based on m/z 349 and m/z 319, so m/z 319 was formed by the loss of one molecule of CH_2_O from m/z 349. Then, m/z 319 went on to lose one molecule of methoxy to form m/z 288, and compound **23** was identified as 8-methyldihydrochelerythrine. The chemical structures and proposed fragmentation pathways of compounds **22** and **23** were shown in Figure 8. Similarly, compound **26** was identified as chelerythrine [24,36]. The chemical structure and proposed fragmentation pathways of compound **26** was shown in Appendix A.

### 2.3. Biological Activity of the MDHW Extract

The results of the UPLC-MS/MS analysis showed that MDHW was dominated by alkaloid components. Alkaloids have been attracting attention for their analgesic, anti-inflammatory, antibacterial, antioxidant, and other pharmacological effects [1], Therefore, in this experiment, the alkaloid fraction of MDHW was enriched and a variety of in vitro bioactivity experiments were performed on the crude extract, alkaloid, and non-alkaloid fractions to evaluate the medicinal value of MDHW, and the results are shown in Table 3.

The experimental results show that all MDHW extracts have certain pharmacological activities, and previous studies also showed that *Magnoliaceae* extracts have good antioxidant, anti-tumor, and other pharmacological activities [10]. The pharmacological activities of the crude extract and alkaloid fraction were significantly stronger than those of the non-alkaloid fraction. The alkaloid fraction was comparable to the crude extract in terms of antioxidant capacity and anti-acetylcholinesterase capacity, and significantly better than the crude extract in terms of anti-α-glucosidase, with an IC_50_ value of 0.04 mg/mL. The results of the present study fill the gap in the pharmacological activity of the MDHW extract and confirmed that alkaloids are the main source of active ingredient.

#### 2.3.1. Antioxidant Properties

Plant materials are excellent sources of natural antioxidants. The antioxidant activity of plant extracts cannot be assessed by a single approach due to the complexity of the phytochemical composition and different mechanisms of antioxidant reactions. Three methods, namely DPPH, ABTS, and the total reducing power, were used to comprehensively evaluate the antioxidant capacity of the MDHW extracts. The analyses of the antioxidant activity of the MDHW extract and the positive control trolox were shown in Figure 9. As shown in Figure 9, the positive control trolox achieves 100% DPPH and ABTS radical scavenging as well as 100% total reducing power before 0.5 mg/mL. The crude extract and alkaloid fractions showed a dose-dependent antioxidant activity between 0.06 and 2 mg/mL, and the non-alkaloid fraction showed a weaker overall antioxidant activity but also showed some dose dependence (0.06–4.00 mg/mL). The DPPH radical scavenging rate: crude extract > alkaloid fraction > non-alkaloid fraction; ABTS radical scavenging rate: crude extract > alkaloid fraction > non-alkaloid fraction; total reducing power: alkaloid > crude extract > non-alkaloid fraction. This shows that the antioxidant capacity of the alkaloid fraction is comparable to or even better than that of the crude extract. It was noteworthy that the total reducing power of the crude extract was weaker than that of the alkaloids, probably due to the antagonistic effect of the components present in the crude extract in terms of total reducing power. It also cannot be excluded that the amount and proportion of each metabolite in the crude extract lead to a decrease in the total reducing power. Similarly to our results, the antioxidant activity of the total alkaloids was comparable to the positive standard a-tocopherol with an ABTS value of 0.5 mg/mL and a FRAP value of 658 mg/100 g [37], and the purified flavonoids were higher than the crude extracts in terms of antioxidant capacity [38].

UPLC-MS/MS analysis revealed that the non-volatile components of MDHW were dominated by isoquinoline alkaloids. Isoquinoline alkaloids have a wide plant distribution and rich chemical structure types, and their unique pharmacological activities are one of the main reasons for interest and research [39]. Aporphine alkaloids were the primary structural type in the current investigation, and prior research indicated that they were a potential class of plant metabolites with a variety of biological and antioxidant activities [40]. Magnoflorine and apomorphine can exert antioxidant activity by scavenging ROS and free radicals and blocking the hypoxanthine–xanthine oxidase system [30]. Liu et al. extracted and isolated 15 aporphine alkaloids, including anonaine, asimilobine, and roemerine, from the lotus flower, and found that all of them had good antioxidant activity by scavenging free radicals, metal complexation, and iron reduction [41]. The antioxidant mechanism of protopine was to boost the activities of catalase, glutathione peroxidase, and superoxide dismutase while inhibiting the growth of intracellular Ca^2+^, the expression of caspase-3, and H_2_O_2_-induced apoptosis [42]. Combining previous studies with our results, it can be speculated that isoquinoline alkaloids are the main contributors to the antioxidant capacity of MDHW extracts.

Overall, the antioxidant capacity of all MDHW extracts was present, proving that they can also be a great source of natural antioxidants. Alkaloids are the active ingredients, the rich isoquinoline alkaloid component may be responsible for the good antioxidant capacity.

#### 2.3.2. Enzyme Inhibition Effects

Acetylcholinesterase (AChE) and α-glucosidase are the key enzymes inhibiting AD and DM. We evaluated the effect of the MDHW extract on the inhibition of these two enzymes, and the results are shown in Figure 10. The positive control tacrine and acarbose reached 100% inhibition at the lowest concentration of 0.06 mg/mL. The crude extract and alkaloid fractions showed dose-dependent enzyme inhibition activity between 0.06 and 2 mg/mL, and the non-alkaloid fraction showed weaker inhibition overall but also showed a certain dose dependence (0.06–4.00 mg/mL). The inhibitory capacity of both enzymes was alkaloid fraction > crude extract > non-alkaloid fraction, indicating that the alkaloid fraction was significantly effective in inhibiting AChE and α-glucosidase. The lower inhibitory effect of the crude extracts than the alkaloid fraction can be explained by complex interactions (synergistic or antagonistic) among phytochemicals.

Many acetylcholinesterase inhibitors were isolated from natural products, and most compounds have been shown to have some inhibitory activity, but alkaloids are considered to be the most promising natural products for the treatment of AD due to their complex N-containing structures [43]. As the structural type of this most abundant compound, aporphine alkaloids not only show excellent activity in antioxidants but also effectively inhibit AChE [40]. Protopine inhibits AChE in a dose-dependent manner and has been shown to be reversible, specific, and competitive [42], and it has also been shown to have almost the same efficacy as velnacrine, the most commonly used tacrine derivative on the market for the treatment of AD [44].

In addition, other alkaloids found in MDHW were also shown to be effective in inhibiting the AChE activity, such as allocryptopine with an IC_50_ value of 250 μM against AChE [45]. Bamatin has a strong inhibitory activity against AChE with an IC_50_ value <10 μM [46]. The inhibitory effect of asimilobine on AChE is comparable to that of the standard galanthamine and is time- and concentration-dependent [44]. Michelalbine has a good binding affinity for potential targets for the treatment of diabetes, namely human amylin peptide and dipeptidyl peptidase-4, and can effectively control blood glucose [47]. In addition, magnoflorine, anonaine, apomorphine, and roemerine also have significant efficacy in insulin resistance and obesity suppression [30,48]. 

Overall, the MDHW extract showed significant effects on AChE and α-glucosidase inhibition, indicating its great developmental value in delaying Alzheimer’s disease and hypoglycemic products. Alkaloids are among the active ingredients, and the isoquinoline alkaloid-rich composition may be responsible for the good AChE and α-glucosidase activity inhibition ability.

#### 2.3.3. Antimicrobial Activities

To investigate the bacterial inhibitory potential of MDHW extracts, two gram-positive bacteria, *Staphylococcus aureus* (*S. aureus*) and *Bacillus subtilis* (*B. subtilis*), as well as two gram-negative bacteria, *Escherichia coli* (*E. coli*) and *Erwinia carotovora* (*E. carotovora*), were subjected to minimum inhibitory concentration (MIC) determination. The results of the MIC determination of the three extracts against the four pathogenic bacteria are shown in Table 4. The positive control 100 ug/mL kanamycin completely inhibited the growth of all strains, while the negative control 50% methanol had no inhibitory ability for all strains. The MIC values of these extracts against the four pathogenic bacteria ranged from 0.125 to 2 mg/mL, indicating that they had a certain inhibitory effect on all the test strains. The enriched alkaloid fraction had stronger bacterial inhibitory effects compared to the crude extract and the non-alkaloid fraction. This may be due to the higher content of alkaloids in the purified extract than in the crude extract, and alkaloids are considered to have antibacterial activity [49]. The non-alkaloid fraction had a weak antibacterial ability with MIC values greater than 1 mg/mL, while the crude extract and alkaloid fraction showed great potential for bacterial inhibition against all tested strains.

The gram-positive and gram-negative bacteria are Class II dangerous biological agents, according to the Advisory Committee on Dangerous Pathogens (ACDP), since they can cause human sickness and perhaps endanger life [7]. These strains are often found in household contaminants such as cereals, meat, eggs, and water contamination, so it is important to inhibit their growth [50]. The microbe most sensitive to all extracts was S. aureus, and the MIC values of crude extract, alkaloid, and non-alkaloid parts were 0.25, 0.125, and 0.5 mg/mL, respectively. All extracts showed stronger inhibition against gram-positive bacteria than gram-negative bacteria. This phenomenon can be explained by the fact that gram-positive bacteria are to some extent more sensitive to drugs than gram-negative bacteria [51].

Many studies have shown that alkaloids, flavonoids, and lignans have antibacterial effects on many pathogenic bacteria [38,52,53]. Combined with the UPLC-MS, the analysis of MDHW showed that MDHW contained effective active antibacterial components such as alkaloids and flavonoids and lignans, especially isoquinoline alkaloids. There are various types of alkaloid structures among which isoquinoline alkaloids and indole alkaloids are the main types of compounds with antibacterial activity [49]. Moreover, 1 μg/mL roemerine significantly inhibited biofilm formation and prevented the transition of *Candida albicans* yeast to mycelium transformation [54]. In previous studies, magnoflorine, palmatine, berberine, allocryptopine, chelidonine, chelerythrine, and anonaine [30,36,53,55] have been shown to have strong antibacterial abilities, and these compounds are found in the present study. In addition to the alkaloids, the flavonoids (kaempferol [56], luteolin [57]) and lignin (podophyllotoxin [58]) found in this study have also been shown to have good antibacterial potential.

The above analysis indicates that MDHW can be a potential source of natural antimicrobial agents, in which alkaloids are the main active ingredient, and the enrichment of isoquinoline alkaloid components may be responsible for the significant antimicrobial activity.

## 3. Materials and Methods

### 3.1. Plant Material

Three plants of *Michelia macclurei* Dandy were collected in May 2021 in Nanning, Guangxi, China, and identified as *Michelia macclurei* Dandy by the Guangxi University Testing Centre. After the removal of the bark and sapwood, the heartwood (Appendix A) was naturally dried, crushed, and filtered through a 40-mesh sieve, and then sealed and dried for storage.

### 3.2. Preparation of Plant Extracts

#### 3.2.1. Volatile Compounds

Steam distillation: extraction of essential oils by SD was carried out according to the literature with slight modifications [59]. Briefly, the volatile oil was extracted by adding 130 g of heartwood powder to 850 mL of water, connecting the volatile oil extractor, and adding 5 mL of ether to the upper layer, heating in an electric heating jacket, and refluxing for 10 h. After cooling, the ether layer was collected to obtain the volatile oil. The volatile oil was dissolved in chromatographic grade n-hexane and analyzed by GC-MS.

Ultrasonic extraction accurately weighs 0.100 g of heartwood powder dissolved in 5 mL of methanol and extracted with ultrasound for 1h. Then, 1 mL of supernatant was evaporated and dissolved in chromatographic grade hexane and passed through a 0.22 μm filter membrane.

#### 3.2.2. Non-Volatile Compounds

Extraction steps were slightly modified from the literature [60]. A total of 100 g of heartwood powder was dissolved in 1000 mL of ethanol (95%, *v/v*) and extracted for 2 h. Extracted twice, the filtrate was concentrated and dried to obtain the crude extract. The crude extract was added to distilled water, adjusted to pH 7.0, and filtered, and the filtrate was the non-alkaloid fraction. The pH of the acid water layer was adjusted to 9.0–10 with 2% NaOH, and chloroform extraction was performed to obtain the chloroform layer and the alkaline water layer. The chloroform layer was recovered and the lipid-soluble alkaloids were obtained. An appropriate amount of n-butanol was added to the alkaline aqueous layer, eluted, and dried to constant weight to obtain water-soluble alkaloids. The lipid-soluble alkaloids and the water-soluble alkaloids were combined to obtain the total alkaloid fraction. All extracts were stored in the refrigerator at −4 °C until use.

### 3.3. GC-MS Analysis

#### 3.3.1. Instrument Condition

The analysis was performed using the Thermo Scientific TRACE 1300-TSQ 9000 chromatograph with the TG-5SILMS column (30 m × 0.25 mm × 0.25 μm), injection port temperature of 280 °C, injection volume of 1 μL, split mode (split ratio 10:1), and a flow rate of 1.0 mL/min. The ramp-up procedure was an initial temperature of 40 °C (hold for 2 min), and ramped up to 280 °C at 5 °C/min (hold for 5 min). Mass spectrometry conditions: scan range 30–700 m/z, solvent delay 3.0 min, transmission line temperature 28 °C, ion source EI, ion source temperature 300 °C, and high purity helium as the carrier gas.

#### 3.3.2. Component Identification

The MS/MS spectra were automatically searched using the Thermo Fisher mass spectrometry database and derived. The retention index RI was calculated in combination with the n-alkane peak exit time. Based on the RI, MS/MS spectra, the latest NIST library to confirm the structure of each chemical component, and the relative percentages of each chemical component were determined using the peak area normalization method.

### 3.4. UPLC-MS Analysis

#### 3.4.1. Instrument Condition

Q-EXACTIVE-MS (Thermo Scientific, Waltham, MA, USA) was used for the analysis. The chromatographic column was ACQUITY UPLCBEHC18 (2.1 mm × 50 mm × 1.7 μm) with a mobile phase of 0.1% formic acid-water in A and acetonitrile in B. The column temperature was 30 °C, the injection volume was 1 μL, and the flow rate was 0.8 mL/min. The column temperature was 30 °C, the injection volume was 1 μL, and the flow rate was 0.8 mL/min. The gradient conditions were 0–3 min, 5–10% B; 3–6 min, 10–30%; 6–15 min, 30–70% B; 15–20 min, 70–95% B; 18–23 min, 95% B; 23–25 min, 95–5% B. The mass spectrometry conditions were electrospray ionization (ESI), an ion source temperature of 300 °C, auxiliary gas flow rate of 69 kPa, scan modes of full MS and full MS/dd-MS^2^, a mass range of 100–1,000 Da, and primary and secondary scan resolutions of 70,000 and 17,500, respectively.

#### 3.4.2. Component Identification

The identification of compounds based on accurate masses, elemental compositions, and secondary mass spectral fragments provided by the mzCloud Mass Spectral Library database. The accurate mass measurements of ions and major characteristic ions were controlled to within ±5.0 ppm of their expected elemental composition.

### 3.5. Biological Activity

The extracts (crude extract, alkaloid, and non-alkaloid fractions) were prepared at concentrations of 0.0625, 0.125, 0.25, 0.5, 1, 2, and 4 mg/mL, and all biological activity tests were carried out using this series of concentrations.

#### 3.5.1. Antioxidant Assays

In this work, three different methods were used to evaluate the antioxidant properties of MDHW extracts. Trolox (for free radical scavenging and the total reducing power) was used as a positive control.

The DPPH assay was carried out according to the method described in the literature with minor modifications [61]. Briefly, a 2.5 mL aliquot of DPPH ethanol solution (0.2 mM) was mixed with the 2.5 mL sample solution. The absorbance was measured at 517 nm after 30 min at room temperature and protected from light.

The ABTS assay was carried out according to the method described in the literature with minor modifications [3]. ABTS (7 mM) and aqueous potassium persulphate (2.45 mM) were mixed in equal volumes and left to stand overnight at room temperature and protected from light to form the ABTS+ solution. The absorbance at 734 nm was adjusted with ethanol to 0.700 ± 0.005 before use. A total of 0.1 mL of the sample solution and 2 mL of ABTS+ solution were mixed. The absorbance at 734 nm was measured after 30 min at room temperature and protected from the light.

The total reducing power determination was carried out according to the method described in the literature with minor modifications [3]. 800 μL of PBS solution (0.2 M, PH 6.6), 1 mL of K_3_Fe (1%) solution, and 100 μL of sample solution were mixed and incubated at 50 °C for 20 min; then, 1 mL of trichloroacetic acid was added (10%), and the mixture was centrifuged and took 1 mL of supernatant; finally, 1 mL of distilled water and 200 μL of FeCl_3_ (0.1%) were added and the absorbance was measured at 700 nm after 10 min.

#### 3.5.2. Enzyme Inhibitory Assays

The AChE inhibition assay was performed according to the method described in the literature with slight modifications [3]. As such, 3 mL of PBS solution (0.1M, PH 8.0), 20 μL AChE solution (0.5 U/mL), and 100 μL sample solution were mixed and incubated for 2 min at 37 °C. The reaction was terminated by the addition of 1 mL of SDS (4%) and the absorbance was quickly measured at 412 nm.

The α-glucosidase inhibition assay was performed according to the literature with slight modifications [3]. As such, 1.5 mL PBS solution (0.1 M, PH 6.8), 200 μL α-glucosidase (0.6 U/mL), and 10 μL sample solution were mixed and incubated for 10 min at 37 °C; then, 160 μL PNPG (2.5 mM) was added and the mixture was incubated for 20 min at 37 °C. The reaction was terminated by the addition of 1 mL of Na_2_CO_3_ (0.2 M) and the absorbance was quickly measured at 405 nm.

#### 3.5.3. Antimicrobial Assay

Four-gram pathogenic bacteria (*S. aureus*, *B. subtilis*, *E. coli*, and *E. carotovora*) were subcultured twice before use. Then, 10^8^ CFU/mL of bacterial broth was prepared by autoclaving a liquid medium (containing 0.9% NaCl). The MIC was determined using a 96-well plate microdilution method with reference to the literature [62]. 50 μL of liquid medium, 50 μL of sample solution, and 50 μL of bacterial solution (10^8^ CFU/mL) were added to each well and incubated for 20 h at 37 °C; then, 20 μL of resazurin solution (0.3 mg/mL) was added and the mixture underwent further incubation for 4 h. The concentration that can cause any color change to the solution was defined as MIC. Kanamycin (100 μg/mL) was used as a positive control and methanol (50%) as a negative control.

### 3.6. Data Analysis

Compound Discoverer 3.1 was used for the mass spectrometry data analysis, and compound structures were drawn using ChemDraw 20.0. Three replicates were set up for the bioactivity experiments and data were reported as the mean ± standard deviation (SD). One-way ANOVA IC_50_ values were calculated using IBM SPSS Statistics 19. Origin 2023 was used for data plotting.

## 4. Conclusions

In this study, a comprehensive analysis of the MDHW extract was conducted for the first time using GC-MS and UPLC-MS. The results show that the MDHW chemical components were of various structural types, with volatile components dominated by sesquiterpenes and their derivatives and fatty compounds, and non-volatile components including alkaloids, lignans, flavonoids, terpenoids, etc., among which isoquinoline alkaloids were the main component types. In vitro bioactivity experiments showed that the MDHW extract has excellent antioxidant, anti-acetylcholinesterase, anti-α-glucosidase activities, as well as antibacterial ability. MDHW can be used as a potential source of bioactive substances that can be developed in natural antioxidants, delay Alzheimer’s disease, hypoglycemic products, and antibacterial agents. Alkaloids are the main active ingredients of MDHW, and the main type of isoquinoline alkaloids may be the main reason for their excellent biological activity. However, the antioxidant inhibiting AD and DM and the antibacterial mechanisms of the alkaloids as well as important polyphenolic compounds in MDHW still need further study. Our research provides helpful information for the further rational exploitation of MD plant resources.

## Figures and Tables

**Figure 1 ijms-24-07972-f001:**
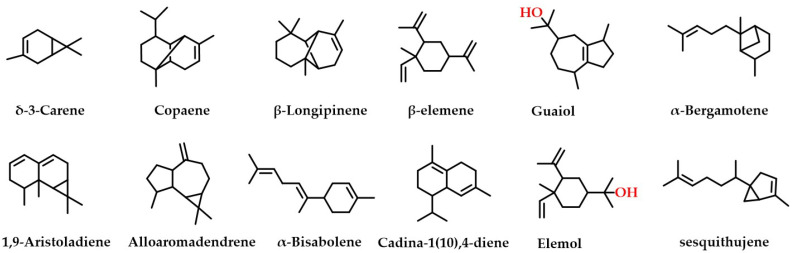
The 12 same components of the two extraction methods.

**Figure 2 ijms-24-07972-f002:**
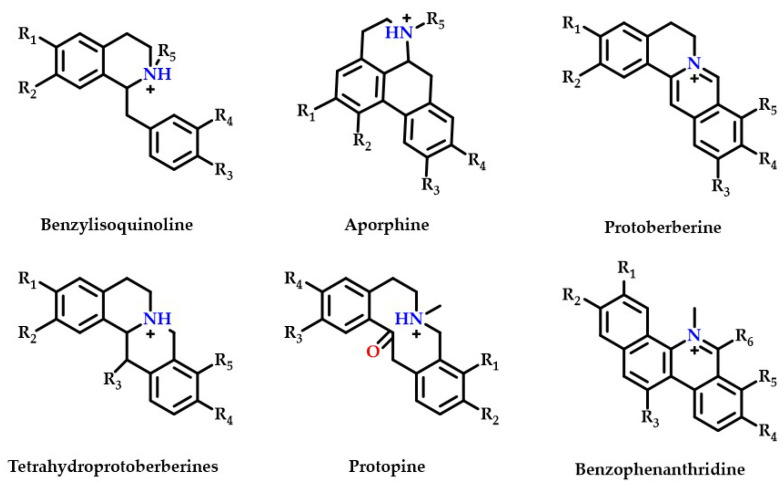
The basic structure of six isoquinoline alkaloids.

**Figure 3 ijms-24-07972-f003:**
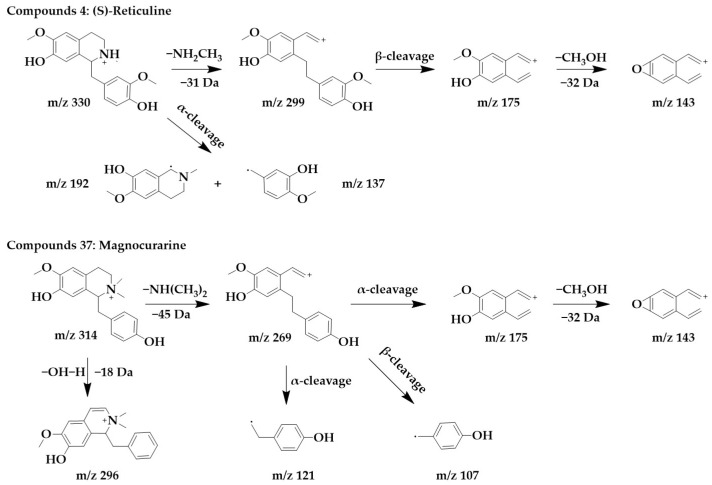
Chemical structure and proposed fragmentation pathways of compounds **4** and **37**.

**Figure 4 ijms-24-07972-f004:**
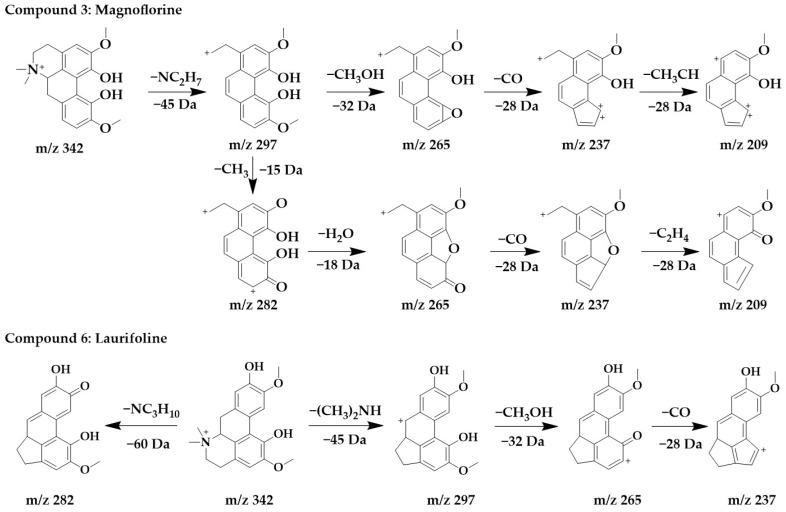
Chemical structure and proposed fragmentation pathways of compounds **3** and **6**.

**Figure 5 ijms-24-07972-f005:**
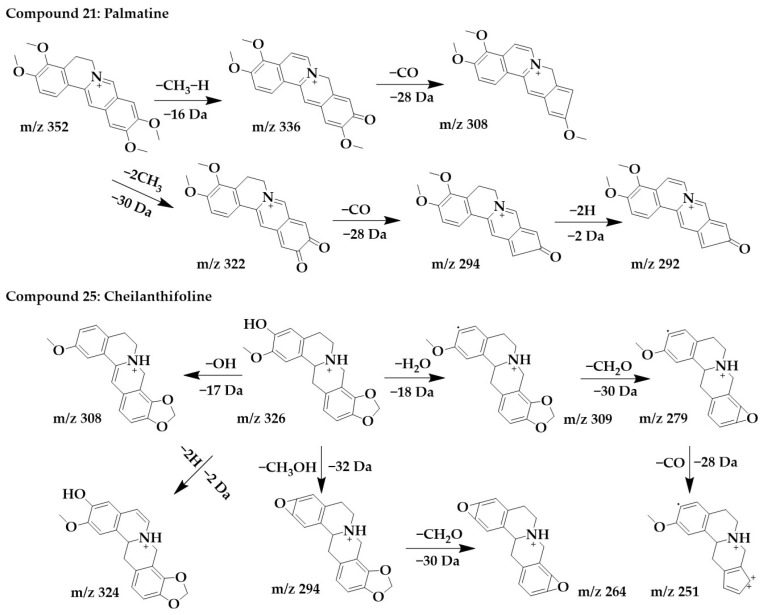
Chemical structure and proposed fragmentation pathways of compounds **21** and **25**.

**Figure 6 ijms-24-07972-f006:**
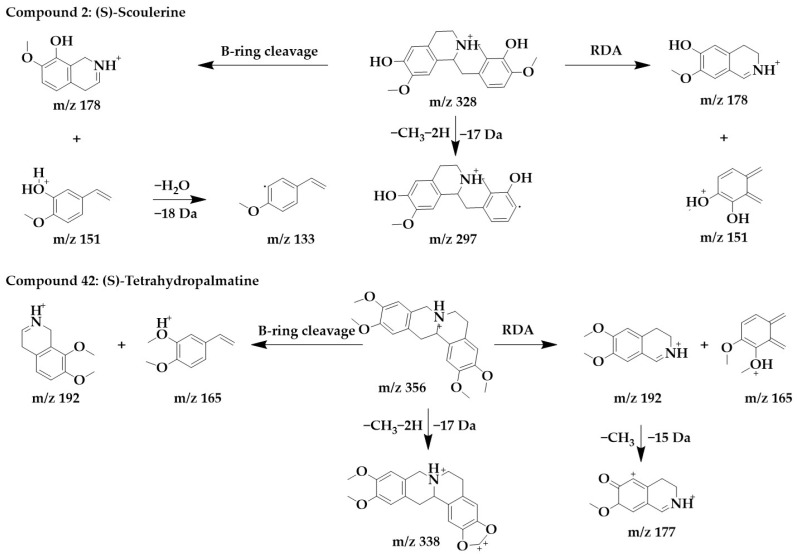
Chemical structure and proposed fragmentation pathways of compounds **2** and **42**.

**Figure 7 ijms-24-07972-f007:**
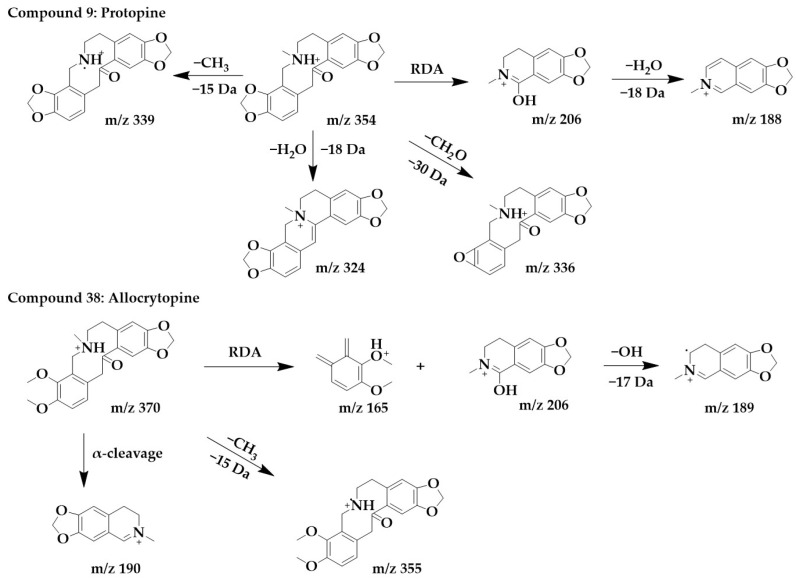
Chemical structure and proposed fragmentation pathways of compounds **9** and **38**.

**Figure 8 ijms-24-07972-f008:**
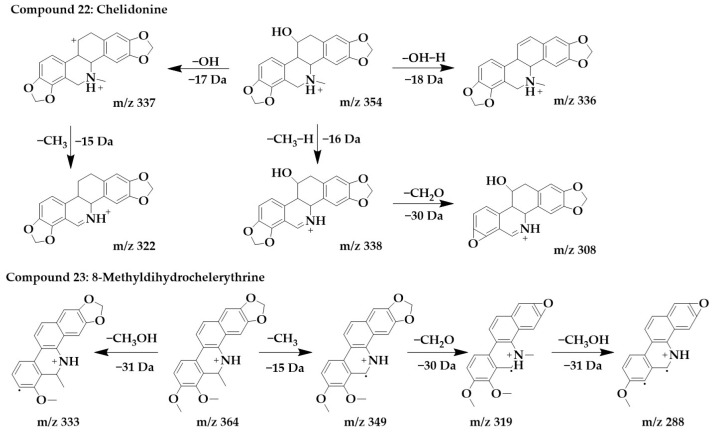
Chemical structure and proposed fragmentation pathways of compounds **22** and **23**.

**Figure 9 ijms-24-07972-f009:**
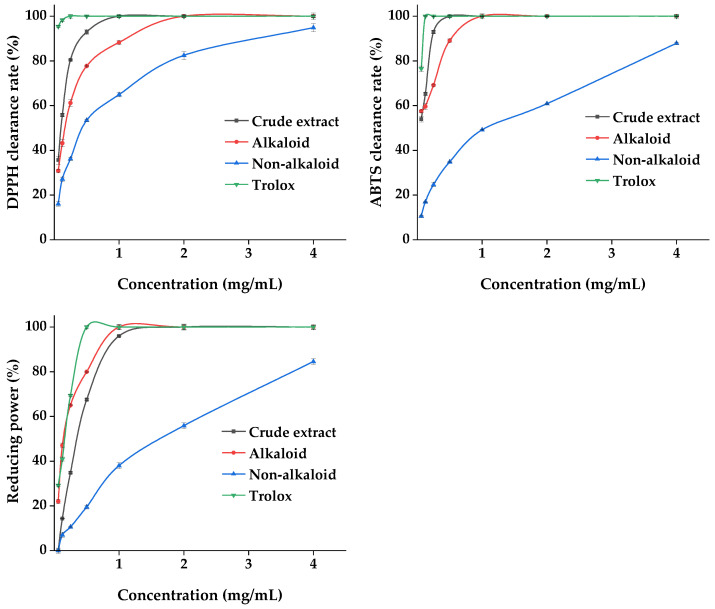
Antioxidant properties of the MDHW extract.

**Figure 10 ijms-24-07972-f010:**
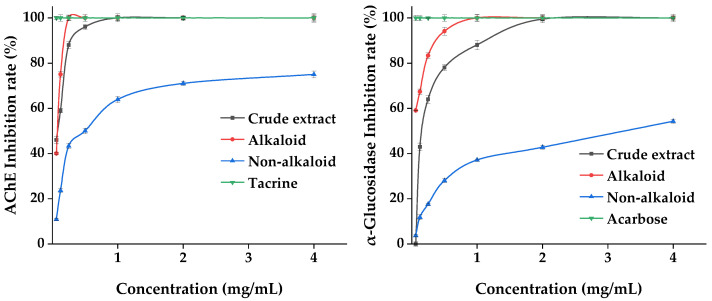
Enzyme Inhibition Effects of the MDHW Extract.

**Table 1 ijms-24-07972-t001:** Part of the volatile components of MDHW obtained by the two extraction methods.

NO	Compounds	RIcal	RIlit	RA (%)		Type	Identification
				SD	UE		
1	δ-3-Carene	1013	1011	0.06	0.07	MH	MS; RI
2	Dodecane, 2,6,11-trimethyl-	1275	1275	-	**1.44**	FC	MS; RI
3	Copaene	1367	1365	0.03	0.51	SH	MS; RI
4	β-Longipinene	1395	1394	0.02	0.19	SH	MS; RI
5	β-elemene	1405	1400	**11.88**	**8.20**	SH	MS; RI
6	sesquithujene	1414	1417	0.42	0.35	SH	MS; RI
7	α-Bergamotene	1422	1422	0.17	0.12	SH	MS; RI
8	caryophyllene	1428	1425	**7.43**	-	SH	MS; RI
9	1,9-Aristoladiene	1441	1435	0.23	0.15	SH	MS; RI
10	Alloaromadendrene	1466	1462	0.30	0.18	SH	MS; RI
11	γ-Muurolene	1474	1472	**2.93**	-	SH	MS; RI
12	α-Bisabolene	1488	1479	1.26	**2.18**	SH	MS; RI
13	β-Bisabolene	1519	1519	**4.25**	-	SH	MS; RI
14	Cadina-1(10),4-diene	1528	1525	0.44	0.44	SH	MS; RI
15	Elemol	1543	1546	2.79	0.56	SO	MS; RI
16	Guaiol	1600	1600	**10.67**	0.54	SO	MS; RI
17	β-Eudesmol	1628	1628	**9.68**	-	SO	MS; RI
18	Epicubenol	1649	1642	**3.92**	-	SO	MS; RI
19	δ-Cadinol	1669	1670	**4.35**		SO	MS; RI
20	β-Bisabolol	1690	1675	**7.14**	-	SO	MS; RI
21	Farnesol	1713	1710	**9.98**	-	SO	MS; RI
22	Crocetane	1805	1813	-	**1.16**	FC	MS; RI
23	Heptadecane,2,6,10,15-tetramethyl-	1908	1914	-	**2.23**	FC	MS; RI
24	7,9-Di-tert-butyl-1-oxaspiro(4,5)deca-6,9-diene-2,8-dione	1921	1916	-	**0.84**	AC	MS; RI
25	Isoeicosane	1952	1962	-	**1.55**	FC	MS; RI
26	Octadecane,3-ethyl-5-(2-ethylbutyl)-	2989	-	-	**1.03**	FC	MS
27	Tris(2,4-di-tert-butylphenyl)phosphite	3401	3396		**29.26**	FC	MS; RI
28	Tris(2,4-di-tert-butylphenyl)phosphate	3580	3582		**18.43**	FC	MS; RI
Group and Count							
	Monoterpene hydrocarbons (MH)			0.78	0.07		
	Sesquiterpene hydrocarbons (SH)			38.74	13.23		
	Oxygenated sesquiterpenes (SO)			57.66	0.54		
	Fatty compounds (FC)			0.00	59.93		
	Aromatic compounds (AC)			0.37	2.38		
	Total identified			97.54	76.15		

Note: “-” indicates that the information is not queried or detected; bold indicates the top ten content; RIcal—retention index calculated by authors; RIlit—retention index by NIST query; RA: relative area; MS: MS/MS spectrum; RI: retention index; grouping and counting were performed for all compounds extracted from SD and UE.

**Table 2 ijms-24-07972-t002:** Non-volatile components of MDHW extracts analyzed by UPLC-MS.

NO	RT (min)	Identification	Molecular Formula	Exact Mass	Error (ppm)	Main Fragments (m/z)
Alkaloid						
1	1.232	Choline	C_5_H_13_NO	103.1001	3.87	88.0238, 84.9603, 72.0814, 60.0817
2 *	1.419	(S)-Scoulerine	C_19_H_21_NO_4_	327.1466	−1.51	297.1110, 179.0896, 178.0862, 151.0752
3 *	1.453	Magnoflorine	C_20_H_23_NO_4_	341.1622	−1.62	297.1119, 282.0880, 265.0859, 237.0910
4 *	1.46	(S)-Reticuline	C_19_H_23_NO_4_	329.1621	−1.75	299.1273, 192.1018, 175.0753, 143.0492
5	1.515	Betaine	C_5_ H_11_NO_2_	117.0579	0.41	72.0451, 59.0738, 58.0659
6 *	1.198	Laurifoline	C_20_H_23_NO_4_	341.1620	−2.12	297.1121, 282.0882, 265.0858, 237.0910
7 *	5.923	Corytuberine	C_19_H_21_NO_4_	327.1460	−3.29	297.1119, 266.0900, 265.0858, 237.0908
8 *	6.025	N,N-Dimethylglaucine	C_21_H_23_NO_5_	369.1562	−3.75	354.1327, 325.1069, 326.1107, 293.0805
9 *	6.179	Protopine	C_20_H_19_NO_5_	353.1258	−1.54	339.1100, 336.2278, 324.0865, 206.1176
10 *	6.204	Bulbocapnine	C_19_H_19_NO_4_	325.11668	−3.07	295.1157, 294.1124, 265. 0851, 237.0548
11 *	6.616	Apomorphine	C_17_H_17_NO_2_	267.1249	−3.87	237.0909, 220.0841,219.0804, 191.0855
12 *	6.924	Norarmepavine	C_18_H_21_NO_3_	299.1515	−2.23	269.1170, 292.1019, 175.0753, 161.0832
13 *	7.005	N-Methylasimilobine	C_18_H_19_NO_2_	281.1043	−3.08	251.1065, 249.0907, 219.0804, 191.0855
14 *	7.143	Isococlaurine	C_17_ H_19_NO_3_	285.1355	−1.50	269.1169, 237.0906, 175.0752, 107.0494
15 *	7.156	(S)-Coclaurine	C_17_ H_19_NO_3_	285.1925	3.99	269.1170, 237.0905, 175.0752, 107.0484
16 *	7.168	Asimilobine	C_17_H_17_NO_2_	267.1249	−3.87	251.1064, 219.0803, 191.0853
17 *	7.183	Ushinsunine	C_18_H_17_NO_3_	295.1198	−3.65	280.1235, 279.1206, 278.1173, 265.0855
18 *	7.748	O-Methylarmepavine	C_20_H_25_NO_3_	327.1822	−3.87	297.1481, 285.1476, 190.0946, 159.0804
19 *	7.498	Corydaline	C_22_ H_27_NO_4_	369.1927	−3.53	206.11754, 192.1012, 179.1605, 165.0908
20 *	8.008	Menisperine	C_21_H_25_NO_4_	355.1043	−3.64	311.1273, 296.1041, 280.1091, 279.1011
21 *	8.070	Palmatine	C_21_H_21_NO_4_	351.1095	−3.47	336.1229, 322.1064, 308.1275, 294.1075
22 *	8.106	Chelidonine	C_20_H_19_NO_5_	353.1249	−3.96	338.1337, 337.1303, 336.1228, 322.1071
23 *	8.722	8-Methyldihydrochelerythrine	C_22_H_21_NO_4_	363.1457	−3.89	349.1306, 333.1312, 319.0916, 288.1016
24 *	8.634	Anolobine	C_17_H_15_NO_3_	281.1615	−4.46	265.1246, 264.1017, 250.0948, 234.1031
25 *	8.819	Cheilanthifoline	C_19_H_19_NO_4_	325.0938	−3.80	324.1592, 309.0950, 308.0915, 294.1157
26 *	9.221	Chelerythrine	C_21_H_17_NO_4_	347.1145	−0.60	333.0993, 318.1170, 302.1170, 290.11835
27 *	10.775	Roemerine	C_18_H_17_NO_2_	279.0887	−3.02	250.0944, 249.0910, 220.0851, 219.0805
28	17.736	(S)-Nicotine	C_10_H_14_N_2_	162.1154	0.39	143.9970, 120.9812, 116.9725, 84.9605
29	17.92	Phenmetrazine	C_11_ H_15_NO	177.1154	0.38	146.9616, 128.9510, 119.0495
30	17.931	Ammodendrine	C_12_ H_20_N_2_O	208.1572	−1.56	167.0123, 162.0073, 143.9968, 84.9603
31 *	18.397	Xanthoplanine	C_21_H_25_NO_4_	355.17747	3.06	311.0907, 296.1039, 280.1065, 279.1009
32 *	18.524	Armepavine	C_19_H_23_NO_3_	313.1670	−2.67	283.1323, 206.1173, 189.0903, 175.0750
33 *	18.527	Michelalbine	C_17_H_15_NO_3_	281.1047	−1.8	265.1123, 264.1018, 250.0949, 234.1038
34 ^*^	18.621	Tetrahydrocolumbamine	C_20_H_23_NO_4_	341.1624	−0.8	323.2523, 192.1020, 165.0912, 151.0755
35	18.673	Acetylcadaverine	C_7_H_16_N_2_O	144.1261	−0.99	121.9664, 123.9644, 72.0451
36	18.707	Acronidine	C_18_H_17_NO_4_	311.1157	−0.18	295.0964, 266.0893, 265.0860, 251.10664
37 ^*^	18.749	Magnocurarine	C_19_ H_23_NO_3_	313.1675	2.18	296.0992, 283.1327, 269.1171, 237.0911
38 *	18.846	Allocrytopine	C_21_H_23_NO_5_	369.1936	0.88	355.1764, 206.1177, 190.0867,165.0909
39 *	18.954	Laudanosine	C_21_H_27_NO_4_	357.1933	−1.87	327.1582, 315.1587, 296.1402,284.1399
40	19.034	Koenigine	C_19_H_19_NO_3_	309.1359	−2.1	269.0859, 253.0813, 252.0777, 207.0799
41 *	19.075	Anonaine	C_17_H_15_NO_2_	265.1102	−0.38	250.0944, 249.0910, 220.0837, 219.0805
42 *	19.1	(S)-Tetrahydropalmatine	C_21_H_25_NO_4_	355.1052	1.85	265.0859, 192.1017, 206.1184, 165.0904
43 *	19.428	Protosinomenine	C_19_H_23_NO_4_	329.1620	−2.11	192.1016, 175.0751, 151.0752, 143.0490
44	19.861	Diethyltoluamide	C_12_H_17_NO	191.1307	−1.44	175.1482, 135.0442, 128.0194, 107.9602
45	20.472	Thalicpureine	C_22_H_27_NO_5_	374.1889	−0.16	340.2593, 322.2490, 312.1594, 295.1329
Flavonoid						
46	1.206	Taxifolin	C_15_H_12_O_7_	304.0577	−1.94	287.0548, 259.0596, 231.0648, 153.0180
47	18.322	Robinin	C_33_H_40_O_19_	740.2144	−2.73	288.0578, 287.0544, 129.0544, 85.0289
48	18.323	Luteolin	C_15_H_10_O_6_	286.047	−2.06	243.0647, 215.0696, 153.0180,149.0231
49	18.427	Kaempferol	C_15_H_10_O_6_	286.0472	−2.05	251.1637, 233.1531, 205.1584, 187.1478
50	18.533	Nictoflorin	C_27_H_30_O_15_	594.1573	−2.01	288.0578, 287.0545, 85.0289, 71.0498
51	18.993	4′,5,6,7-Tetramethoxyflavanone	C_19_H_20_O_6_	344.1252	−2.28	327.1231, 299.0921, 267.1016, 177.0548
52	19.126	Rhoifolin	C_27_H_30_O_14_	578.1617	−3.16	409.0904, 287.0547, 271.0593, 127.0390
53	19.26	Kievitone	C_20_ H_20_O_6_	356.1252	−2.21	321.1117, 307.0958, 165.0543, 137.0596
54	20.485	Dimefline	C_20_H_21_NO_3_	323.1519	−0.75	279.8514, 266.1257, 233.0962, 163.1115
Lignan						
55	18.922	Podophyllotoxin	C_22_H_22_O_8_	414.1305	−2.39	235.0962, 195.0650, 181.0493, 89.0602
56	19.212	Acetoxypinoresinol	C_22_H_24_O_8_	416.1462	−2.31	233.0801, 191.0698, 181.0491, 167.0700
57	19.37	Matricin	C_17_H_22_O_5_	306.1458	−3.1	265.1432, 247.1326, 229.20, 201.1270
58	20.377	Magnoshinin	C_24_H_30_O_6_	414.2034	−2.02	120.0888, 119.0856
59	21.344	Cinchonain Ia	C_24_H_20_O_9_	452.1101	−1.33	322.2477, 239.1487, 133.0859, 89.0602
60	22.14	Peiminine	C_27_H_43_NO_3_	429.3236	−1.56	177.1122, 133.0860, 123.1171, 89.0602
Terpenoid						
61	19.22	Gibberellin A7	C_19_H_22_O_5_	330.1465	−0.71	285.1120, 255.1016, 223.0755, 151.0754
62	21.067	Samandarin	C_19_H_31_NO_2_	305.2352	−0.83	260.8843, 250.1800, 194.1177, 142.0561
63	22.538	5-Androstenetriol	C_19_H_30_O_3_	306.2191	−1.18	233.1536, 187.1481, 147.1170, 123.1170
Coumarin						
64	19.517	Pteryxin	C_21_H_22_O_7_	386.1362	−0.93	357.1337, 353.1117, 340.2560, 137.0599
65	19.873	7-Hydroxycoumarine	C_9_H_6_O_3_	162.0316	−0.41	135.0442, 133.0285, 107.0858, 89.0603
Lipid						
66	19.103	12-Aminododecanoic acid	C_12_H_25_NO_2_	215.1886	0.32	198.1853, 156.1747, 155.0702
67	23.613	Macamide B	C_23_ H_39_NO	345.3027	−1.28	239.2373, 137.1325, 133.0859, 91.0548

Note: “*” indicates that the compound is an isoquinoline alkaloid.

**Table 3 ijms-24-07972-t003:** In vitro bioactivity IC_50_ values of MDHW extracts.

Sample	IC_50_(mg/mL)				
	DPPH	ABTS	Total Reducing Power	AChE	α-Glucosidase
Crude extract	0.10 ± 0.001 ^a^	0.05 ± 0.034 ^a^	0.29 ± 0.006 ^a^	0.07 ± 0.015 ^a^	0.21 ± 0.051 ^a^
Alkaloid	0.16 ± 0.026 ^a^	0.05 ± 0.002 ^a^	0.14 ± 0.004 ^b^	0.08 ± 0.019 ^a^	0.04 ± 0.008 ^b^
Non-alkaloid	0.42 ± 0.041 ^b^	0.84 ± 0.020 ^b^	1.43 ± 0.023 ^c^	0.56 ± 0.036 ^b^	2.21 ± 0.032 ^c^

Note: Values are expressed as means ± SD. Data marked with different letters indicate a significant difference (*p* < 0.05).

**Table 4 ijms-24-07972-t004:** The MIC of MDHW extracts against four pathogenic bacteria.

Samples	*S. aureus*	*B. subtilis*	*E. coli*	*E. carotovora*
Crude extract (mg/mL)	0.25	0.25	1	0.5
Alkaloid (mg/mL)	0.125	0.25	0.5	0.25
Non-alkaloid (mg/mL)	0.5	1	2	2
Kanamycin (100 μg/mL)	-	-	-	-
Methanol (50%)	+	+	+	+

Note: “-” indicates that the solution shows blue color, indicating that 100 ug/mL kanamycin completely inhibits the growth of bacteria; “+” indicates that the solution shows red color, indicating that ethanol cannot inhibit bacteria.

## Data Availability

Data are available from the corresponding author on request.

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
