# Peer review of "A Study of the Chemical Composition and Biological Activity of Michelia macclurei Dandy Heartwood: New Sources of Natural Antioxidants, Enzyme Inhibitors and Bacterial Inhibitors"

_ijms, 2023, doi:10.3390/ijms24097972_

Round 1
Reviewer 1 Report
This work is devoted to a comprehensive study of Magnolia wood (Michelia macclurei). The authors use extraction methods to determine the component composition of raw materials. The article is generally well written. The main idea is clear. Despite a number of obvious advantages of this work, I recommend finalizing the following points:
1. Title. From the title of the article, one might get the impression that the authors are exploring the wood itself, and not the extracts.
2. All abbreviations must be deciphered at the beginning of the article.
3. It is desirable to compare the results obtained in more detail with those known from the literature (including closely related plants).
4. It is desirable to unify the drawings.
5. If a scheme of chemical transformations taken from the literature is given, then it is desirable to indicate the source.
6. The presence of what substances in the studied extracts determines the biological activity?
7. Please cite: 10.3390/molecules27186129.
8. : It is desirable to indicate the composition of the original wood.
9. It is desirable to indicate in more detail the preparation of wood. Was the bark removed before grinding? The components of bark are somewhat different from those of wood.
Author Response
We appreciate for Reviewers' warm work earnestly, and have revised the content according to your suggestions. Please see the attachment. We hope that the correction will meet with approval.

Reviewer 2 Report
1) in table 1 group the compounds by chemical family.
2) some news references were added in the revized paper:
a) DOI: 10.3390/metabo13030371: metabolite (2023)
b) DOI: 10.1016/j.jff.2022.105327: journal of functional foods (2022)
c) DOI: 10.3390/life12101571: life (2022)
3) some questions arise for the authors:
a) What is the main question addressed by the research?
b) Do you consider the topic original or relevant in the field? Does it address a specific gap in the field?
c) What does it add to the subject area compared with other published material?
Author Response

(The authors gave the same response as above.)

Reviewer 3 Report
In the submitted manuscript Chen et al. investigated the chemical composition and biological activities of the heartwood of Michelia macclurei Dandy (MDHW) and analyzed volatile and non-volatile chemical components of MDHW by using GC-MS and UPLC-MS, respectively. To determine in vitro bioactivities and primary bioactive compounds of MDHW, the antioxidant, enzyme inhibition, and antibacterial properties of the crude extract, alkaloid, and non-alkaloid fractions were also assessed.
The subject of this manuscript is relevant to the field. However, there are a few comments that should be addressed before the final publication of the manuscript:
Introduction
The new Figure of Michelia macclurei Dandy (MDHW) would be very beneficial for the readers.
Section Results and Discussion
Subsection 2.1. Volatile Components of MDHW Determined with GC-MS
Lines 123-125: The references addressing the use β-elemene and β-eudesmol as clinical compounds should be provided.
Figure 1: I advise the authors to change the color of the chemical structures from red to black. The oxygen atoms should remain red.
Subsection 2.2. Nonvolatile Components of MDHW with UPLC-MS/MS
Figure 2: I advise the authors to change the color of the chemical structures of isoquinoline alkaloids from red to black. The nitrogen atoms should be blue.
Subsection 2.2.1. Benzylisoquinoline alkaloids
A new figure with chemical structure and proposed fragmentation pathways of compound 37 (magnocurarine) should be provided in the main text.
Subsection 2.2.3. Protoberberine alkaloids
A new figure with chemical structure and proposed fragmentation pathways of compound 25 (cheilanthifoline) should be provided in the main text.
Subsection 2.2.4. Tetrahydroprotoberberine alkaloids
A new figure with chemical structure and proposed fragmentation pathways of compounds 42 ((S)-tetrahydropalmatine) should be provided in the main text.
Lines 278-281: The authors should mention that two figures with chemical structures and proposed fragmentation pathways of compounds 19 and 34 are present in the Supplementary materials.
Subsection 2.2.5. Protopine alkaloids
Line 296: The number 39 in the name »compound 39« should be revised to 38.
A new figure with chemical structure and proposed fragmentation pathways of compound 38 (allocrytopine) should be provided in the main text.
Subsection 2.2.6. Benzophenanthridine alkaloids
Two figures with chemical structures and proposed fragmentation pathways of compounds 23 (8-methyldihydrochelerythrine) and 26 (chelerythrine) should be provided in the main text.
Subsection 2.3 Biological activity of MDHW extract
Table 3: The corresponding controls should be listed in the table footnote. The designation of significance level p < 0.05 with three different letters »a,b,c« should be revised to »*«.
Subsection 2.3.3. Antimicrobial activities
The authors state that MDHW contains various compounds (alkaloids, flavonoids, and lignans) with confirmed antioxidant and antibacterial activity. The potent antioxidant and antimicrobial activities of polyphenols (flavonoids) from MDWH should also be discussed.
3. Materials and Methods
3.5 Biological activity
The applied protocols of DPPH, ABTS, reducing power, enzyme inhibition assays, and antimicrobial activity assays should be briefly described.
Conclusions
Line 566: The insights into inhibitory mechanisms of alkaloids as well as important polyphenolic compounds from MDHW on antioxidative, and antimicrobial proteins as well as on proteins involved in Alzheimer's disease progression can be revealed through advanced molecular dynamics techniques and free-energy calculations as well as through inverse molecular docking. Advanced supercritical fluid extraction techniques could also be applied to obtain MD extracts rich in valuable polyphenolic compounds.
References:
1. Pantiora, P.; Furlan, V.; Matiadis, D.; Mavroidi, B.; Perperopoulou, F.; Papageorgiou, A.C.; Sagnou, M.; Bren, U.; Pelecanou, M.; Labrou, N.E. Monocarbonyl Curcumin Analogues as Potent Inhibitors against Human Glutathione Transferase P1-1. Antioxidants 2023, 12, 63. https://doi.org/10.3390/antiox12010063
2. Kores, K.; Kolenc, Z.; Furlan, V.; Bren, U. Inverse Molecular Docking Elucidating the Anticarcinogenic Potential of the Hop Natural Product Xanthohumol and Its Metabolites. Foods 2022, 11, 1253. https://doi.org/10.3390/foods11091253
3. Furlan, V.; Bren, U. Helichrysum italicum: From Extraction, Distillation, and Encapsulation Techniques to Beneficial Health Effects. Foods 2023, 12, 802.
Line 146: The sentence »..fragmentation pathways of compounds of the compounds were analyzed.« should be revised. The phrase »of compounds« is doubled and should be removed.
Author Response

(The authors gave the same response as above.)
